# Adaptations of Potential Nitrogenase Activity and Microbiota with Long-Term Application of Manure Compost to Paddy Soil

**Zhalaga Ao [1], Juan Xia [2], Honoka Seino [3], Katsuhiro Inaba [2], Yukitsugu Takahashi [4], Chie Hayakawa [3], Hideaki Hirai [3] and Isamu Maeda [1,2,*]**

1   Department of Applied Life Science, United Graduate School of Agricultural Science, Fuchu Campus, Tokyo University of Agriculture and Technology, Fuchu 183-8509, Japan

2   Department of Applied Biological Chemistry, School of Agriculture, Mine Campus, Utsunomiya University, Utsunomiya 321-8505, Japan

3   Department of Agrobiology and Bioresources, School of Agriculture, Mine Campus, Utsunomiya University, Utsunomiya 321-8505, Japan

4   Faculty of Agriculture, Utsunomiya University, Moka 321-4415, Japan

\*   Correspondence: i-maeda@cc.utsunomiya-u.ac.jp; Tel.: +81-28-649-5477

**Abstract:** Biological nitrogen fixation complements nitrogen from fertilizers in crop plants under natural conditions. It also contributes to the reduction in chemical fertilizer (CF) utilization in cultivated lands, which fits the concept of sustainable agriculture. From this viewpoint, however, it is still unknown in paddy fields how soil bacterial nitrogenase and microbiota are affected by applied materials in the soil. Therefore, in this study, the effects of long-term material applications on potential nitrogenase activity and the microbiota of soil bacteria were investigated. The nitrogenase activity tended to be higher in manure compost (MC)-applied soils than in CF-applied soils in both summer and winter. Soil bacterial alpha diversity increased whereas soil ammonia availability decreased with the MC application. The dynamic response of soil bacterial microbiota was caused by the MC application. The abundance of *Nitrospira*, a class of ammonia and nitrite oxidation bacteria, was lower and the abundance of alpha-Proteobacteria was higher in the MC-applied soils than in the CF-applied soils. These results suggest that the alpha diversity increase and restricted availability of $NH_3$-N might contribute to the increase in potential nitrogenase activity in the long-term MC-applied soils.

**Keywords:** paddy field; soil; rice; bacterial microbiota; nitrogen fixation





## 1. Introduction

Ammonia is the main source of nitrogen (N) for the production of synthetic N fertilizers. Ammonia is synthesized through the Haber–Bosch process at typical temperatures of around 500 ºC and pressures of around 20 MPa in the chemical industry [1]. Hydrogen as an integral reactant for the Haber–Bosch process is mostly refined from natural gas by steam methane reformation, which produces greenhouse gas emissions [2]. Through the replacement of N fertility based on biological resources with synthetic N fertilizers, a sizable percentage of the human population depends on the use of fossil fuels for survival [3]. Although some countries are fundamentally dependent on synthetic N fertilizers for food production, obtaining N from biological sources is more sustainable than from industrial sources. Many countries have the potential to reduce their dependence on synthetic N fertilizers.

Biological N fixation (BNF) is catalyzed by nitrogenases of diazotrophic bacteria, and the biosynthesis of ammonia proceeds using atmospheric N as a substrate under natural environmental conditions [4]. The most important N-fixing agents in agricultural systems are the symbiotic associations between crop and forage/fodder legumes and diazotrophs symbiosing inside root nodules of legumes [5]. Carbon (C) sources such as malic acid, fumaric acid, and succinic acid are supplied through plant glycolysis to nodule

bacteroids, which utilize these organic acids for the energy to fuel nitrogenase activity [6]. For chemolithoheterotrophic bacteria, C sources also have fundamental roles as energy sources and electron donors, which are required for the nitrogenase reaction [7]. Besides nodule bacteroids inside root nodules, diazotrophs, such as rhizobia and methanotrophs living at sites where C and energy sources are predominantly available, are responsible for BNF in non-legume plants [8]. Diazotrophic bacteria can exist freely or in symbiosis in agricultural lands. BNF in agronomically significant non-legume plants and the plant-associative diazotrophs have been reported in sugarcane stems, stems and storage tubers of sweet potato, and aerial root-associative mucilage of maize [9].

In paddy rice fields, methane is produced from organic compounds excreted from the root, and BNF by methane-oxidizing methanotrophs takes place in the root tissues and also in the surface soil [10]. Low-molecular-weight organic acids and alcohols formed through microbial degradation of rice straw are suitable carbon sources for the growth of purple non-sulfur bacteria, whose BNF in paddy fields is known [11]. The application of molybdenum (Mo), which is required for the nitrogenase FeMo-cofactor, greatly increased BNF in Mo-deficient paddy fields and stimulated the growth of the N-fixing cyanobacteria [12]. An elevated $CO_2$ concentration increased the abundance *nifH*, which encodes dinitrogenase reductase of diazotrophs, in rice roots and rhizosphere soils with N fertilization at the rice tillering stage [13]. These studies have shown the importance of the rhizosphere and root and the impact of environmental conditions on BNF and bacterial microbiota formation. However, it is still unknown how soil bacterial microbiota and the population of diazotrophs in the microbiota are affected, especially with the long-term application of agricultural materials to paddy fields.

It has been reported that the application of manure compost (MC) to paddy fields increases the alpha diversity of soil microbiota [14]. Therefore, cow MC for N fertilization may also function as a source of diazotrophic bacteria in the soil. In addition, the presence of excess mineral N in the soil inhibits both nodule formation and nitrogenase activity [15]. It has been reported that soil microbial populations are generally C-limited and organic C application stimulates soil microbial growth until it is eventually limited by available N [16]. When the C/N values of 33 in bamboo sheaths, 58 in bamboo twigs [17], and 64.68 in bamboo sawdust [18] were compared with the C/N value of 20.02 in cow manure [19], it was supposed that bamboo powder (BP) application might increase C/N in soil. If C sources degraded from BP might be supplied to soil bacteria, it was conceivable that N demand and charges of energy and reducing power would increase in the soil bacteria. Furthermore, the shift of growth limitation by C sources to growth limitation by N sources might direct the soil bacterial population to the preferential growth of diazotrophs.

In this study, therefore, the effects of long-term applications of MC and CF in combination with BP on potential nitrogenase activity and microbiota in paddy soil bacteria were investigated. The advantages of long-term MC application for BNF in paddy fields were discussed based on the nitrogenase activity, soil ammonia availability, bacterial alpha and beta diversities, and relative taxonomic abundance.

## 2. Materials and Methods

### 2.1. Rice Culture and Sampling

A rice plant (*Oryza sativa* ssp. *japonica*) was cultured on the research farm at Utsunomiya University (Moka, Tochigi, Japan). CF (Japan agricultural cooperatives) and cow MC prepared in the research farm have been separately applied to 1980-$m^2$ allophanic andosol fields, which are divided into 11 plots (180 $m^2$), since 1991 (Figure S1). BP (Midori building, Motegi, Tochigi, Japan) was applied to plots of the CF- and MC-applied fields in 2018 and 2019 and a plot of the MC-applied field in 2020 (Table 1). The long-term CF- and MC-applied fields were irrigated from a single water source. Plots for CF, CF + BP, MC, and MC + BP were fixed over the experimental periods. Puddling work and irrigation were carried out and then rice seedlings were planted in May. Grains were harvested in late September.

**Table 1.** C and N contents of materials and their application quantities and months.

| Applied Material | Content (% *w/w*) | Applied Quantity | Applied Month |
|---|---|---|---|
| Basal CF Additional CF MC BP | N, 10 N, 17 N, 2.7 C, 27.1 N, 0.21 C, 48.6 | 3 kg N/1000 m$^2$ 3 kg N/1000 m$^2$ 27 kg N/1000 m$^2$ 50 kg/1000 m$^2$ | May July March May (2018) in MC plot December (2018) in CF plot March (2019) in MC plot December (2019) in CF plot March (2020) in MC plot |

Soil was sampled before material application and plantation (March, April), after material application (May), at stages of tillering (June) and panicle initiation (July), and after harvest (November, February) as indicated below. Soil was collected at 3 to 5 spots per 180 m$^2$ plot with at least 10 m intervals. When sampling was performed after the rice plantation, a point between rice stocks was selected for sampling. Soil in the middle of the 0–15 cm layer was collected using a 100 mL cylindrical soil sampler with a height of 5 cm and stored in a plastic bag at 4 °C.

### 2.2. Reagents Used

Special-grade reagents were used for media preparation and measurements. N and hydrogen for gas chromatography and acetylene for nitrogenase activity measurement were supplied from compressed gas cylinders. Ethylene (99.5%) and ethane (99.5%) spray cans were purchased from GL Sciences (Tokyo, Japan).

### 2.3. Analyses of Chemical Composition

Mineralizable N in soil was measured by the microbiological assay method [20]. Soil equivalent to 14 g of dry weightand distilled water were poured into a test tube. The test tube was capped with a rubber stopper and incubated at 30 °C for 4, 8, or 12 wk. Ammonia in soil was dissolved in 2 M potassium chloride solution by shaking the test tube for 30 min and extracted using the steam-distillation method. Extracted ammonia was titrated with sulfuric acid solution. Ammonia contents before and after microbiological incubation at 30 °C were defined as ammonia and mineralizable N in soil, respectively.

Total C and N contents of dried cow MC and BP were measured with an NC analyzer Sumigraph NC-22F (Sumika chemical analysis service, Osaka, Japan).

### 2.4. Diazotroph Population Density

For colony formation, bacterial cells in soil were diazotrophically cultured on modified bromothymol blue medium containing 0.5% (*w/v*) L-malic acid [21] and Brown's medium containing 0.5% (*w/v*) glucose [22]. In the modified bromothymol blue medium, glycerol was omitted, and 20 mg/L Bacto yeast extract (Becton, Dickinson company, Sparks, MD, USA) was added. After serial dilutions, soil suspension was spread onto the solid surfaces of modified bromothymol blue medium and Brown's medium. The Petri dish after inoculation was sealed with Parafilm (Bemis, WI, USA), and incubated at 30 °C for 7–9 d. Colony-forming unit (CFU) per g wet weight soil was calculated from the number of colonies formed on both media as a diazotroph population density in soil.

### 2.5. Measurement of Potential Nitrogenase Activity

Soil bacterial nitrogenase activity was measured using 10 g wet weight soil by the acetylene reduction assay in the presence of glucose as an energy and electron source [23]. Soil and 0.4% (*w/w*) glucose dissolved in 4 mL water were poured into a 100 mL Erlenmeyer flask. After covering the flask with a double cap (Kokugo, Tokyo, Japan), 5 mL acetylene

was injected through a needle and syringe. Then, the flasks were sealed with plastic tape and incubated at 30 °C for 24 h.

After incubation, 100 µL of the gas phase in the flask was taken by Gastight syringe 81,000 (Hamilton, Reno, NV, USA), and the ethylene formed was quantified using a gas chromatograph GC-4000 (GL Sciences) equipped with a flame ionization detector and a capillary column Rt-Alumina BOND/KCl (30 m, 0.32 mm ID, 5 µm df, Restek, Tokyo, Japan). N was used as a carrier gas at a flow rate of 3 mL/min. The injector and column temperatures were kept at 200 °C and 100 °C, respectively.

### 2.6. Evaluation of Soil Bacterial Microbiota

DNA in soil was extracted using ISOIL for Beads Beating (Nippon gene, Tokyo, Japan). Portions of soil placed in the bead tubes were subjected to bead beater treatment using a beads crusher µT-12 (Taitec, Koshigaya, Japan). The construction of the metagenomic library targeting the 16S rRNA gene V3 and V4 regions and sequencing by a next-generation sequencer MiSeq (Illumina, San Diego, CA, USA) were performed with an NGS service by Macrogen (Tokyo, Japan) or Bioengineering Lab (Sagamihara, Japan) as described previously [24]. The primer sequences were trimmed, pair sequence reads were merged, and the merged reads comprising 400–470 nucleotides were extracted using Geneious Prime ver. 2019.2.3 (Tomy Digital Biology, Tokyo, Japan). Sequence data were analyzed by mothur (v. 1.47.0) [25]. Measurements of alpha diversity, beta diversity, and relative taxonomic abundance were performed according to the procedure described previously [26]. The sequences were classified with the Greengenes reference database.

## 3. Results

### 3.1. Effects of the Seasonal Change and the Different Material Applications on the Diazotroph Population Density and the Potential Nitrogenase Activity

The diazotroph population density based on CFU in both media was higher in July (at the panicle initiation stage) than in April (before plantation) and November (after harvest) in 2018 (Figure 1). When the population density was compared within the same sampling month, it tended to increase in MC- and MC + BP-applied soils in November.

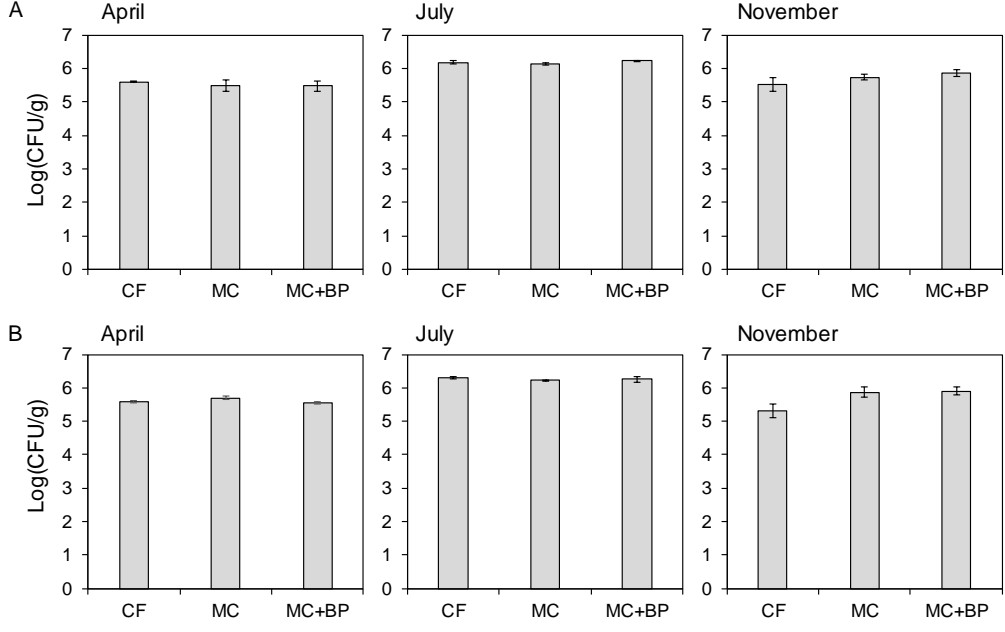

**Figure 1.** Diazotroph population densities in soils collected from CF, MC, and MC + BP plots. CFUs on modified bromothymol blue medium (**A**) and Brown's medium (**B**) were calculated. A column with an error bar indicates a mean ± SD.

The potential nitrogenase activity was higher in July than in April and November in 2018 regardless of the materials applied (Figure 2A). When the nitrogenase activity was compared within the same sampling month, it tended to increase in MC- and MC + BP-applied soil in July and November. Only trace activities were detected in the soils without the addition of glucose (Data not shown). These results indicate that diazotrophs having potential nitrogenase activity were abundant in MC- and MC + BP-applied soil in July and November while in situ BNF in paddy soil might be restricted by the available C concentrations even in the presence of BP. In order to confirm the effects of additional BP application on the paddy soils having higher BNF potentials, the potential nitrogenase activity was measured using CF- and MC-applied soils with and without BP application in July 2019. The nitrogenase activity was higher in MC- and MC + BP-applied soils than in CF- and CF + BP-applied soils (Figure 2B). Then, soil after harvest and rice straw plow (SP) was sampled in February 2023. It was considered that the soil environments after harvest and SP might be less influenced by the materials that were applied before and during rice cultivation, compared to the soil environments at the panicle initiation stage. The potential nitrogenase activity was also higher in MC- and MC + BP-applied soils than in CF-applied soil and no fertilizer (NF) soils with and without SP even after harvest and SP (data not shown). No significant effect of BP application on nitrogenase activity was observed in the soils collected in 2018, 2019, and 2023. These results suggest that the MC application, compared to the CF application, would increase the population density of soil diazotrophs that could exhibit nitrogenase activity at the rice panicle initiation stage and after harvest.

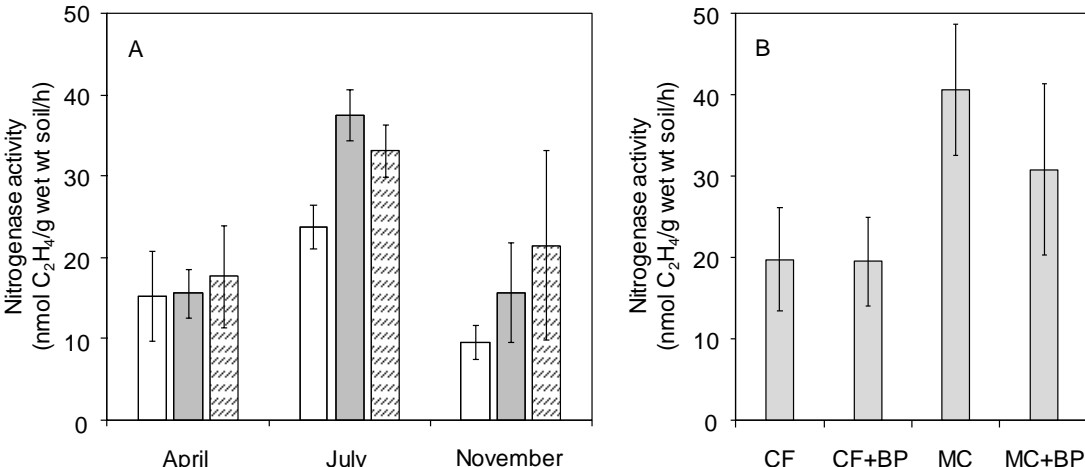

**Figure 2.** Effects of sampling months and different material applications on the potential nitrogenase activity in soil bacteria. The nitrogenase activity was measured using soils collected from CF (open columns), MC (gray columns), and MC + BP (hatched columns) plots in 2018 (**A**) and July 2019 (**B**). A column with an error bar indicates a mean ± SD.

Soil ammonia and mineralizable N were measured in soils collected in 2019. The ammonia concentrations were higher in March regardless of the materials applied (Table 2). The ammonia concentrations continued to decrease in May and July whereas the mineralizable N concentrations increased in May and then decreased in July regardless of the materials applied. Especially, the lower ammonia availability was found in the MC- and MC + BP-applied soils in May and July.

**Table 2.** Ammonia and mineralizable N contents in soils.

| Material Applied | Ammonia (mg/100 g) | | | Mineralizable N (mg NH₃/100 g) | | |
|---|---|---|---|---|---|---|
| | March (Before application) | May (After application) | July (Before additional application) | March (Before application) | May (After application) | July (Before additional application) |
| CF | 2.83 ± 0.43 | 1.93 ± 2.37 | 0.75 ± 0.38 | 3.15 ± 2.61 | 7.06 ± 5.01 | 5.09 ± 1.09 |
| CF + BP | 2.94 ± 1.25 | 1.64 ± 1.66 | 0.70 ± 0.25 | 2.89 ± 1.33 | 6.03 ± 4.86 | 5.35 ± 1.52 |
| MC | 4.21 ± 1.07 | 0.88 ± 0.25 | 0.75 ± 0.33 | 3.18 ± 3.42 | 8.74 ± 1.69 | 7.16 ± 0.61 |
| MC + BP | 2.91 ± 1.20 | 0.93 ± 0.21 | 0.70 ± 0.36 | 2.44 ± 3.28 | 10.27 ± 2.47 | 7.84 ± 1.79 |

*3.2. Increases in Alpha Diversity of Soil Microbiota with the Long-Term MC Application*

The numbers of observed OTUs with given sequence numbers were plotted in soil bacterial microbiota of July 2019. The rarefaction curves of microbiota in MC-applied soils steeply rose with the increase in sequence number compared to those in CF-applied soils (Figure 3). The BP application to the MC-applied field decreased the observed OTU numbers with given sequence numbers.

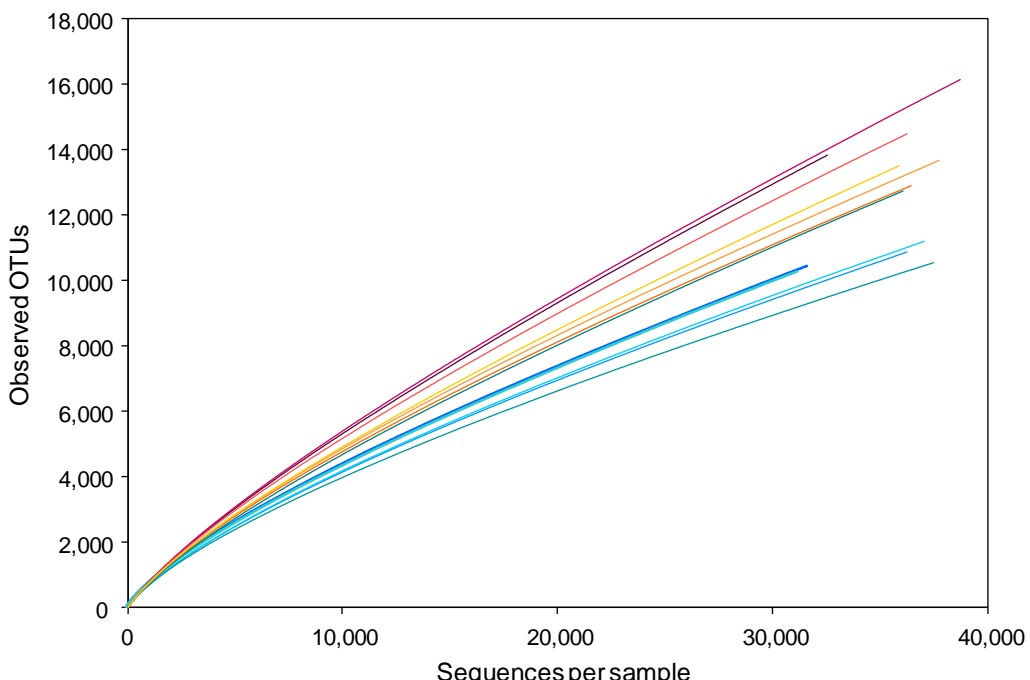

**Figure 3.** Rarefaction curves of OTUs observed in the soil bacterial microbiota in July 2019. Curves for CF (blue), CF + BP (green), MC (red), and MC + BP (yellow-orange) microbiota in different sampling spots are depicted.

Alpha diversity within a bacterial community was measured with Shannon and inverse Simpson indices. The higher values for both Shannon and inverse Simpson indices were observed in MC- and MC + BP-applied soils than in CF- and CF + BP-applied soils (Table 3). These results indicate that the long-term MC application increased the soil bacterial diversity within a bacterial community and the additional BP application to the MC-applied field decreased the soil bacterial diversity at the rice panicle initiation stage.

**Table 3.** Shannon and inverse Simpson indices of the soil bacterial microbiota in July 2019.

| Material Applied | Shannon | Inverse Simpson |
|---|---|---|
| CF | 8.12 ± 0.05 | 717 ± 63 |
| CF + BP | 8.17 ± 0.18 | 757 ± 101 |
| MC | 8.70 ± 0.07 | 1170 ± 37 |
| MC + BP | 8.48 ± 0.03 | 899 ± 154 |

### 3.3. Soil Microbiota Beta Diversity with the Long-Term MC and CF Applications

Beta diversity among the soil bacterial microbiota in July 2019 were compared with the principal coordinate analysis (PCoA). The dissimilarity of bacterial microbiota was observed between MC- and CF-applied soils whereas the effects of the additional BP application to the MC- and CF-applied fields on beta diversity were limited (Figure 4). These results suggest that the long-term different material applications affected the soil bacterial microbiota at the panicle initiation stage.

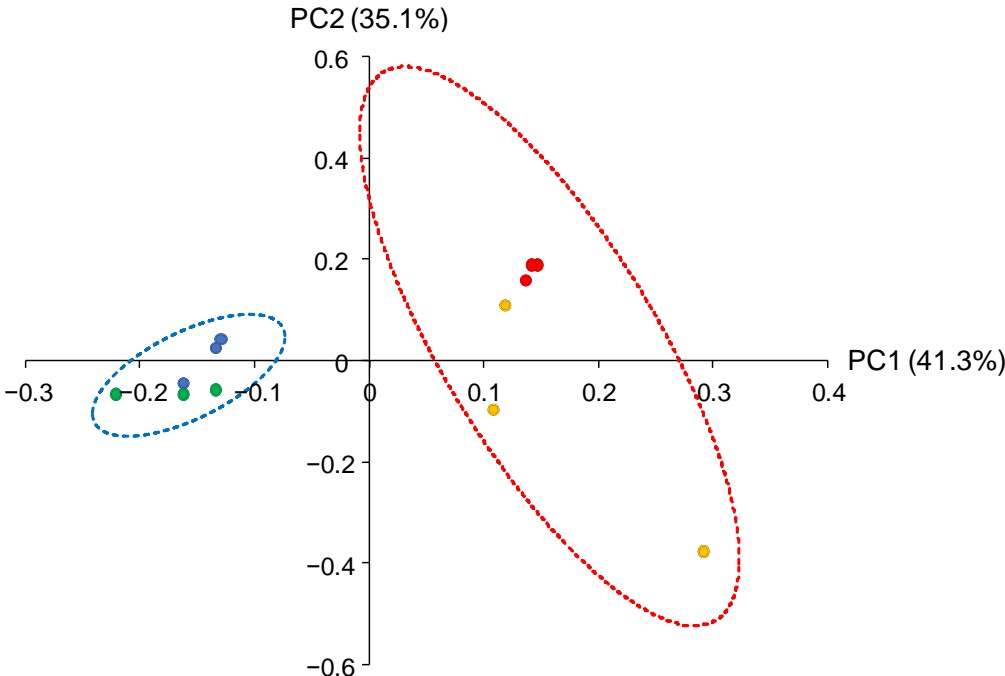

**Figure 4.** PCoA of the soil bacterial microbiota in July 2019. Coordinate points for CF (blue), CF + BP (green), MC (red), and MC + BP (yellow) microbiota are depicted. The dotted drawings show 95% confidence ellipses for CF and CF + BP (blue) and MC and MC + BP (red).

PCoA was also performed in long-term MC- and CF-applied soil in February 2023. At the same time, the bacterial microbiota was also investigated in NF soils with and without SP. The CF and MC applications caused differences in the soil bacterial microbiota in winter (Figure 5). The coordinate points for microbiota in the CF-applied soil were localized within the coordinate point distribution of microbiota in the NF and NF + SP soils. On the other hand, the coordinate points for microbiota in the MC-applied soil were partially placed beyond the coordinate point distribution of microbiota in the NF and NF + SP soils. The long-term MC application affected the soil bacterial microbiota even though rice was not cultivated in winter. The effects of SP in the NF soils and BP application in the MC-applied soils were not observed in winter soil.

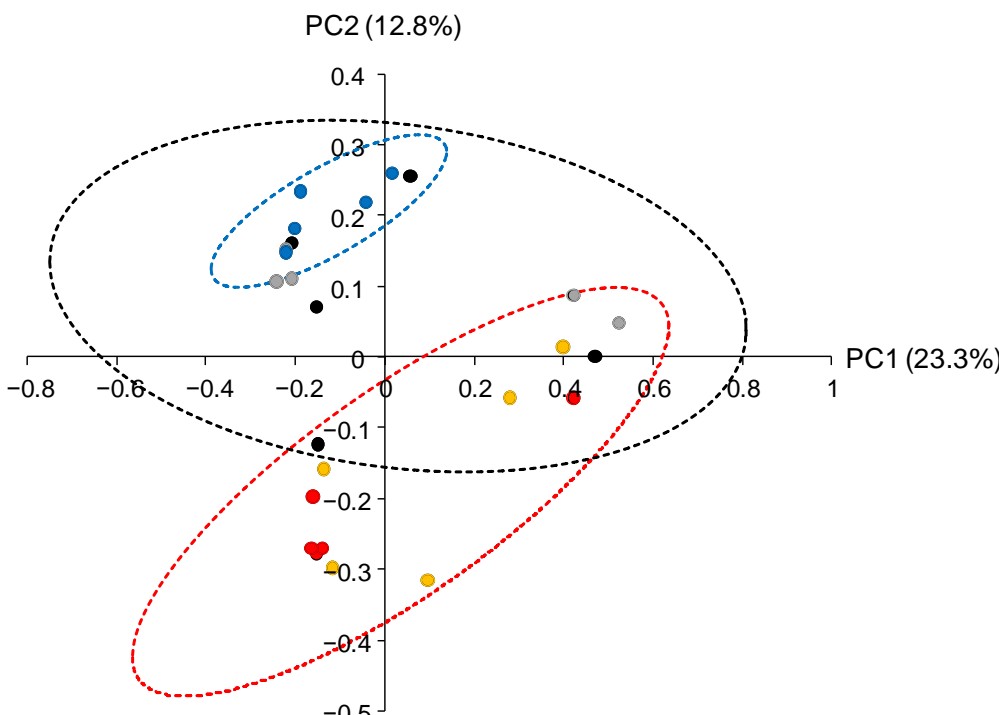

**Figure 5.** PCoA of the soil bacterial microbiota in February 2023. Coordinate points for CF (blue), MC (red), MC + BP (yellow), NF (black), and NF + SP (gray) microbiota are depicted. The dotted drawings show 95% confidence ellipses for CF (blue), MC and MC + BP (red), and NF and NF + SP (black).

*3.4. Changes of the Relative Taxonomic Abundances in the Long-Term MC Application*

The differences in the relative taxonomic abundances were compared with the soil bacterial microbiota observed in CF- and MC-applied fields in July 2019 and February 2023. The phylum abundance of *Nitrospirota* was lower in the microbiota of MC and MC + BP soils than in those of the CF and CF + BP soils in July (Figure 6A). In the microbiota of the MC soil, the abundance of *Chlorobi* decreased, whereas the abundance of *Chloroflexi* and *Actinobacteria* increased compared to those in the CF, CF + BP, and MC + BP soils. These tendencies in the phylum abundances of *Nitrospirota*, *Chlorobi*, *Chloroflexi*, and *Actinobacteria* were observed in the microbiota of CF- and MC-applied fields in February (Figure 6B).

Marked differences in the relative abundances of alpha-Proteobacteria (Figure 7A) and *Nitrospira* (Figure 7B) classes were observed in CF- and MC-applied fields in July. In the class alpha-Proteobacteria, the abundance was higher in the MC microbiota than in the CF and CF + BP microbiota. The MC microbiota had a lower abundance of *Nitrospira* than those in the CF and CF + BP microbiota. The abundance of *Nitrospira* also decreased in the MC + BP plot compared to those in the CF and CF + BP plots. Increases in *Anaerolineae* and delta-Proteobacteria and decreases in beta-Proteobacteria, *Acidobacteria*, and BSV26 detected in the anoxic rice paddy soil [27] were observed in the MC microbiota compared to the CF + BP microbiota. The MC and MC + BP microbiota had lower abundances of gamma-Proteobacteria than those in the CF and CF + BP microbiota. The class abundance also tended to increase in alpha-Proteobacteria (Figure 7C) and decreased in *Nitrospira* (Figure 7D) in the microbiota of the MC-applied field in February as observed in those in July.

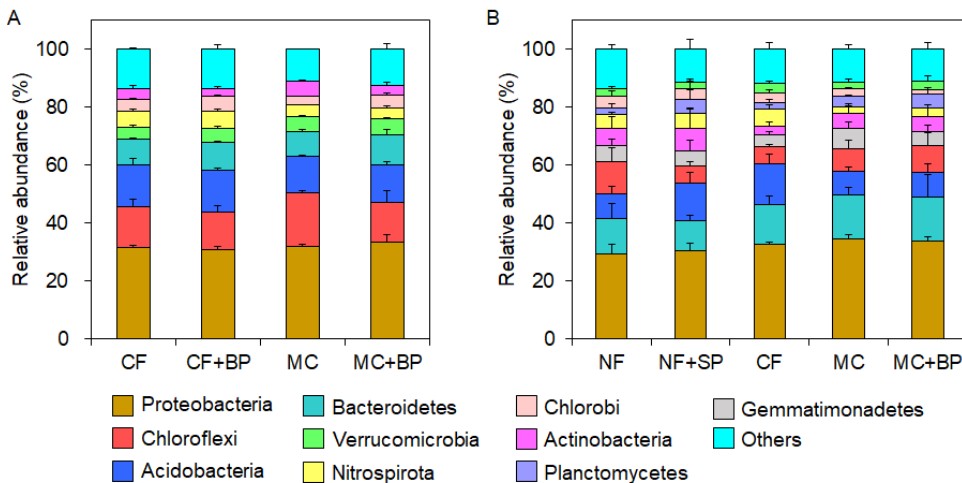

**Figure 6.** Relative phylum abundances in the soil bacterial microbiota. Stacked column charts for the microbiota in July 2019 (**A**) and February 2023 (**B**) are shown. A column with an error bar indicates a mean ± SD.

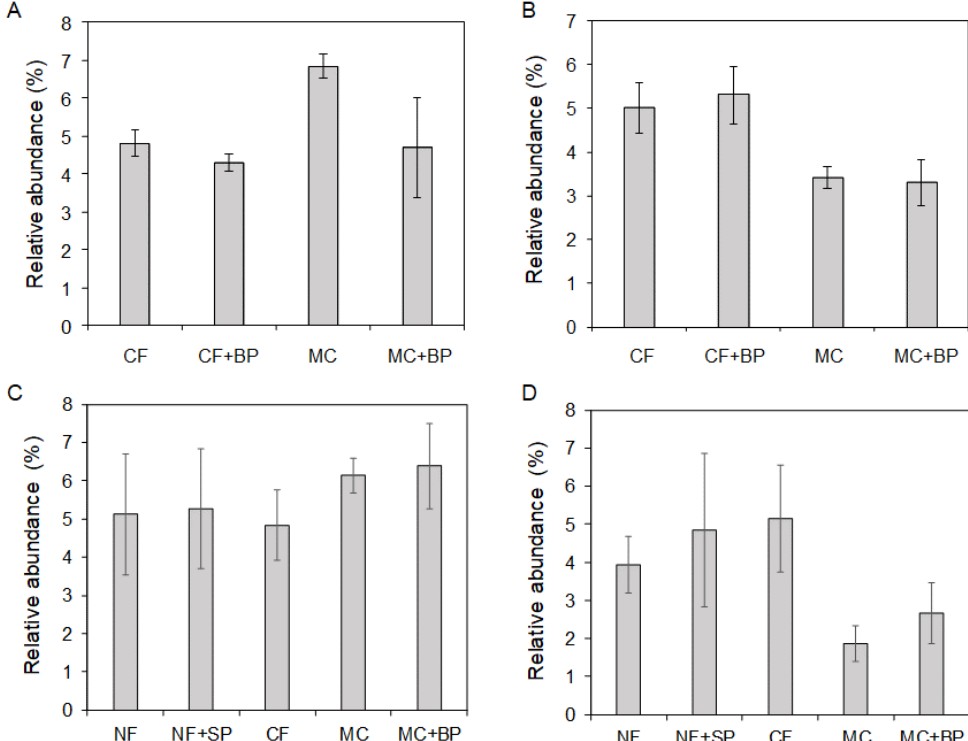

**Figure 7.** Relative class abundances in the soil bacterial microbiota. Column charts of relative abundances for alpha-Proteobacteria (**A,C**) and *Nitrospira* (**B,D**) classes are shown in the microbiota in July 2019 (**A,B**) and February 2023 (**C,D**). A column with an error bar indicates a mean ± SD.

## 4. Discussion

This study indicated that the diazotroph population density and the potential nitrogenase activity in paddy soil increased at the flooded rice-culturing stage compared to those before plantation and after harvest. It was also conceivable that diazotrophs having the potential nitrogenase activity in the soil would increase with the long-term MC application compared to the long-term CF application. This result is supported by the report, in which the higher *nifH* gene copy numbers in the CF + manure treatments in June compared to the NF and CF treatments have been shown [28]. On the other hand, it is likely that the

effect of the MC application and the additional BP application on in situ BNF might be limited. This could be explained by a hypothesis that neither the MC application nor the BP application would connect to the formation of a sufficient amount of available energy and electron sources in paddy soil. Endophytic and rhizosphere bacteria in rice plants have been considered to partially originate from the outside environment [29]. Therefore, the long-term MC application to paddy soil might be effective in increasing BNF by such endophytic and rhizosphere bacteria that could more easily acquire energy and electron sources from rice plants.

Long-term MC application increased the potential nitrogenase activity in paddy soil. One possible reason for the change could be explained by the limited availability of ammonia in the MC-applied soil. In July, before additional CF application, lower ammonia concentrations were measured in the CF- and MC-applied soils whereas in May, lower ammonia concentrations were measured only in the MC-applied soils. On the other hand, total body, straw, and grain weights in the rice cultures at the CF- and MC-applied paddy fields were almost equivalent (Table S1). This suggests that larger N complementation through endophytic and rhizosphere bacterial BNF could be estimated in the MC-applied field. Such restricted $NH_3$–N availability in soil may be advantageous to the growth of diazotrophs.

It has been reported that manure composts enhance the amount and diversity of microorganisms, microbial biomass, soil respiration, and enzyme activities [30,31]. In this study, long-term MC application increased the alpha diversity in soil bacterial microbiota. On the other hand, the additional BP application to the MC-applied field decreased the alpha diversity at the rice panicle initiation stage. This tendency of alpha diversity changes seemed to correlate with the tendency of nitrogenase activity changes in response to the different material applications. Therefore, together with the effect of N availability limitation, continuous inputs of bacterial species and organic matter from MC into paddy soil might contribute to the abundance of diazotrophs that were responsible for the potential nitrogenase activity in soil. This is supported by the preceded study, in which the long-term manure fertilizer treatment influenced the soil microbial community composition and physiology [32].

The long-term different material applications affected the soil beta diversity. It has been shown that the application of CFs did not influence any type of microbial communities in soil [30]. In this study, this tendency could be reproduced in the paddy soil in winter, where CF has been applied since 1991. In addition, the long-term CF application caused a decrease in bacterial diversity in soil [33]. These reports, together with the result of this study, have indicated no effect or negative effect of CF application to the crop field soil on the formation of a diverse group of organisms. Therefore, the difference in the bacterial community structure between the MC- and CF-applied soils might be regarded as a dynamic feature of the soil bacterial microbiota in response to the MC application.

In rice plants, N-fixing nodule formation as found in legume symbiosis does not take place naturally [34]. The rice root–flooded soil interface has been proposed as the N-fixing site and heterotrophic diazotrophs such as *Azotobacter* and *Clostridia* are thought to be responsible for BNF under dark while utilizing root secretions of carbonaceous origin at the rhizosphere [9]. The paddy soil with long-term MC application might be a suitable source of diazotrophs for the BNF at the rhizosphere. The marked differences in phylum abundance between the microbiota of MC- and CF-applied soils were found in *Nitrospirota*, *Chlorobi*, *Chloroflexi*, and *Actinobacteria*. The decreased abundance of *Chlorobi* has been reported in the long-term 30% organic manure application to the paddy field [35]. The growth of *Chlorobi* might be inhibited by the higher organic C content under the application of organic manure. It is probable that the abundance differences at the phylum and class taxa between the microbiota of MC and CF or CF + BP would be also caused by the higher organic C content through MC application. The phylum *Nitrospirota* and the class *Nitrospira* were abundant in the microbiota of CF and CF + BP soils compared to those of MC and MC + BP soils. The result could be explained by the reports, which have indicated that the

*Nitrospira* community shift was affected by pH, $NH_4^+$, and moisture [36] and a high rate of organic manure amendment suppressed the growth of *Nitrospirota* by 20–35% in the paddy fields [37]. It has been found that decreased amounts of soil chemical properties such as $NH_3$–N decreases the abundance of anaerobic ammonium oxidizers when CFs are applied long-term in paddy soils compared with upland soils in Bangladesh [38]. The phylum *Nitrospirota* comprises six sublineages consisting of canonical nitrite oxidizers and complete ammonia oxidizers [39]. Therefore, the decreases in the phylum *Nitrospirota* and the class *Nitrospira* might be caused by the paddy soil fertility such as the restricted availability of $NH_3$-N. It might also result in the preferential growth of diazotrophs in the MC-applied fields. In this study, the alpha-Proteobacteria abundance increased in the microbiota of MC-applied soil. It has been shown that soil *nifH* gene sequences are dominated by alpha-, beta-, and delta-Proteobacteria as well as by Cyanobacteria, and the relative abundance of alpha-Proteobacteria is lower under high N in the paddy soil [40]. The report has also indicated that N-fixing bacteria are abundant in soils with high biodiversity. Hence, the increases in the potential nitrogenase activity by soil bacteria, the soil bacterial alpha diversity, and the specific bacterial abundances such as alpha-Proteobacteria might be beneficial to the BNF at the rice root–flooded soil interface.

## 5. Conclusions

Long-term MC application to the paddy soil resulted in increases in the potential nitrogenase activity and the bacterial alpha diversity in the soil at the rice panicle initiation stage, compared to those in the long-term CF application. On the other hand, it decreased ammonia availability in the soil. Obvious decreases in the phylum *Nitrospirota* and the class *Nitrospira* were detected in the microbiota of MC- and MC + BP-applied soils, suggesting the lower availability of $NH_3$-N in the MC-applied field. A potential BNF contributor alpha-Proteobacteria was abundantly detected in the microbiota of the MC-applied field. As the interface between rhizosphere soil and root has been considered as a BNF site of rice, the long-term MC application might participate in the complementation of N source in rice via supplying diazotrophs in addition to N materials. The accuracy of preliminary conclusions in this study would be confirmed by further analyses in rice plants and rhizosphere soil.

**Supplementary Materials:** The following supporting information can be downloaded at: https://www.mdpi.com/article/10.3390/environments10060103/s1, Figure S1: Schematic representation of the allophanic andosol fields with irrigation system; Table S1: Weights of total body, straw, unfilled rice grain, and brown rice grain.

**Author Contributions:** Conceptualization, I.M., H.H., C.H. and Y.T.; methodology, I.M., H.H., C.H. and Y.T.; software, I.M.; formal analysis, Z.A., J.X., H.S. and K.I.; resources, H.H., C.H. and Y.T.; data curation, Z.A.; writing—original draft preparation, I.M. and Z.A.; writing—review and editing, I.M.; funding acquisition, I.M. and H.H. All authors have read and agreed to the published version of the manuscript.

**Funding:** This work was supported by a Grant-in-Aid for Scientific Research (C) from the Japan Society for the Promotion of Science (17K07709).

**Data Availability Statement:** The raw data associated with this study were deposited into the DNA Data Bank of Japan (DDBJ), BioProject database (accession: PRJDB14253).

**Acknowledgments:** This work was carried out based on management in paddy rice production by stuffs of Utsunomiya University Farm.

**Conflicts of Interest:** The authors declare no conflict of interest.

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
