# Peer review of "Adaptations of Potential Nitrogenase Activity and Microbiota with Long-Term Application of Manure Compost to Paddy Soil"

_environments, doi:10.3390/environments10060103_

Round 1

Reviewer 1 Report (Previous Reviewer 1)

This is the third version of the article and, although the authors have included a new sampling (which, by the way, has not been described in Material and Methods), the quality of the work does not vary significantly. In fact, the results generated by that new sampling are far from conclusive, as it tends to blur the differences observed in the samples corresponding to 2019.

Another aspect to note is the absence in this new version of statistical treatment of the data. Although it could be argued that, in relation to the massive sequencing data, such analysis could be considered as having been performed with the PCoAs, there is no analysis of any kind in relation to the results concerning cultivables or nitrogen compounds.

Finally, no results are to be included in Introduction, and in Figure 3 they have retained nomenclature from previous versions, and the meaning of each sample is not specified in the legend.

In the abstract, "An abundance of..." should be replaced by "The abundance of..."

Author Response

Thank you very much for your valuable time for checking and reviewing the manuscript. Our responses to the comments and suggestions are listed as below.

Reviewer 1’s comments

Another aspect to note is the absence in this new version of statistical treatment of the data.

Authors’ response

Statistical treatment of the data has been removed based on the comments by reviewer 1 (too descriptive) and reviewer 2.

Reviewer 1’s comments

Although it could be argued that, in relation to the massive sequencing data, such analysis could be considered as having been performed with the PCoAs, there is no analysis of any kind in relation to the results concerning cultivables or nitrogen compounds.

Authors’ response

We are thinking that the importance of findings in this research is discussed based on the soil density of viable diazotrophs, potential nitrogenase activity, and soil N availability, together with 16S rRNA gene sequencing data.

Reviewer 1’s comments

Finally, no results are to be included in Introduction, and in Figure 3 they have retained nomenclature from previous versions, and the meaning of each sample is not specified in the legend.

Authors’ response

Description explaining the result has been replaced with a sentences explaining what we did in this study. In figure 3, the meaning of each sample has been added to the legend. We did not add data from the additional experiment in figure 3.

Reviewer 1’s comments on the Quality of English Language

In the abstract, "An abundance of..." should be replaced by "The abundance of..."

Authors’ response

“An abundance of...” has been replaced with “The abundance of...”.

Reviewer 2 Report (Previous Reviewer 2)

Comments:

1) I consider that the manuscript has been improved in the sense that it includes new data and presents the results in a descriptive way and discusses the trends observed in the data without relying on the results of statistical tests that the sample design does not allow.

2) I am still missing a paragraph at the end of the conclusions that makes the real scope of the conclusions explicit, more or less like this:

“Notwithstanding, the conclusions of this study should be taken as preliminary until confirmed by further experiments.”

Author Response

Thank you very much for your valuable time for reviewing the manuscript. Our responses to the comments and suggestions are listed as below.

Reviewer 2’s comments:

1) I consider that the manuscript has been improved in the sense that it includes new data and presents the results in a descriptive way and discusses the trends observed in the data without relying on the results of statistical tests that the sample design does not allow.

Authors’ response

We could confirm the tendency of the relative taxonomic abundance changes and the potential nitrogenase activity increase in the manure-applied soil by the additional experiments.

Reviewer 2’s comments:

2) I am still missing a paragraph at the end of the conclusions that makes the real scope of the conclusions explicit, more or less like this:

“Notwithstanding, the conclusions of this study should be taken as preliminary until confirmed by further experiments.”

Authors’ response

The sentence “The accuracy of preliminary conclusions in this study would be confirmed by further analyses in rice plant and rhizosphere soil” has been added in Conclusions.

This manuscript is a resubmission of an earlier submission. The following is a list of the peer review reports and author responses from that submission.

Round 1

Reviewer 1 Report

After the first round of review, the authors have changed the focus of the article. They have limited the information provided to soils, eliminating that concerning rice cultivation, and have focused mainly on the study of microbiota, including the calculation of diversity indices. However, the modifications made do not improve the quality of the work. In spite of separating the Results and Discussion sections, which should have a positive effect on the depth of the analysis carried out, the study presented here continues to be superficial and mostly descriptive. In the Results section, comments already present in the first version have been left out, which, conceptually, should not be in a section in which no explanations are to be provided. There are even conceptual errors, such as the consideration that beta diversity may increase in certain treatments. Beta diversity considers levels of similarity between samples, i.e., it establishes comparisons, so it can hardly be said to increase.

Regarding the experimental design, the microbial study is carried out with samples corresponding to July 2019, when the soils treated with compost had received two incorporations of this material and, on the other hand, those treated with chemical fertilizers only one incorporation or, in the best case, the second one has just been added, so it could hardly have an impact on the microbiota. On the other hand, and considering that Figure 3 shows three values for each treatment, why is there no standard deviation in Table 3?

On the other hand, there are some expressions that sound, to say the least, strange. Some examples:

- The term microbiota is not used in the plural.

- L. 245-246: "Differences in the relative taxonomic abundances were compared with the soil bacterial microbiotas observed in CF- and MC-applied fields in July 2019". Relative abundances are not compared with the microbiota. What is done is to compare the relative abundances in the different microbial communities associated with each sample.

- The expression a class or a phylum is used on numerous occasions. They should not have indefinite articles, since there is only one class Nitrospira or one phylum Proteobacteria, so the class Nitrospira or the phylum Proteobacteria should be written.

Reviewer 2 Report

After reading the revised version, I find that the fundamental flaw in the experimental design that I pointed out in the first review seems more masked than resolved. Section 2.1 is now very difficult (or impossible) to understand in reference to a "180m2 plot" and "180-plots" not mentioned in the original version.

On the other hand, the same statistical techniques and signification levels continue to be used in the same way, and the same conclusions are drawn, when irremediable flaws in the experimental design do not allow it.

In my opinion, the article cannot be published in its current form. 

The only way I can think of it could be posted would be:

1) eliminating the reference to the parametric test and significance in the methodology, i.e "Multiple comparisons between means were performed"

2) Eliminate the “significant” word when it refers to the difference between means (e.g., “columns with different letters indicate different means”).

3) Replace expressions as “demonstrate” (or similar) by “suggest” o similar.

4) Include an explicit paragraph in the discussion (and, perhaps, some indication in the summary and conclusions) indicating that it is an exploratory study whose results must be confirmed by a larger ongoing experiment, which includes a greater number of plots and repetitions.

I believe that this would be the only way to situate the results obtained in their real context: a descriptive and preliminary exploratory study prior to a rigorous subsequent study that will make it possible to confirm (or not) their results.

Reviewer 3 Report

I recommend publishing it in it's present form.